

# Ateliers de Style
## System rezerwacji wizyt w salonie fryzjerskim

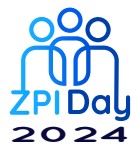

**Autorzy**: Daniel Niezgoda ⬤ · Bartosz Strzecha ⬤ · Jakub Susz ⬤ · Remigiusz Szalek ⬤
**Opiekun:** Hanna Mazur

#### Streszczenie

Celem niniejszego projektu było zaprojektowanie, implementacja i przetestowanie kompleksowego systemu rezerwacji wizyt w salonie fryzjerskim, na który składają się aplikacja mobilna, webowa i zintegrowany z nimi serwer API. Projekt został wykonany z myślą o klientach pewnej sieci salonów fryzjerskich, umożliwiając im przeglądanie oferty usługowej sieci, umawianie wizyt w salonach, zarządzanie nimi oraz ich ocenę, a także udział w programie lojalnościowym. System pozwala również zarządzać swoimi wizytami pracownikom salonów, a właścicielowi - informacjami na temat działalności salonów oraz umożliwia mu generowanie raportów. Zespół projektowy z sukcesem zrealizował wszystkie zaplanowane prace, dostarczając działające aplikacje spełniające wymagania funkcjonalne i techniczne. Zaimplementowane rozwiązanie nie tylko upraszcza proces rezerwacji, lecz także wspiera rozwój relacji z klientami i podnosi poziom satysfakcji z usług. Projekt kończy się pełną gotowością systemu do wdrożenia, co świadczy o jego stabilności, wydajności oraz zdolności do wspierania rozwoju sieci salonów fryzjerskich.

## 1  WPROWADZENIE

Współczesne salony fryzjerskie coraz częściej stają przed wyzwaniem zapewnienia klientom wygodnego i szybkiego dostępu do swoich usług. Tradycyjne metody rezerwacji, takie jak rozmowy telefoniczne czy osobista wizyta w salonie, nie spełniają już oczekiwań konsumentów, którzy preferują rozwiązania cyfrowe. Chociaż na rynku istnieją już systemy rezerwacji wizyt, ich główną wadą są wysokie prowizje pobierane od umawianych usług, co generuje dodatkowe koszty dla przedsiębiorców. Ponadto większość dostępnych rozwiązań nie oferuje opcji programów lojalnościowych, która mogłaby przyciągać klientów oraz wzmacniać ich długoterminowe zaangażowanie. Wspomniane kwestie ograniczają potencjał salonów do budowania relacji z klientami oraz do tworzenia własnych, zindywidualizowanych strategii promocji.

Celem projektu było wyjście naprzeciw powyższym problemom i opracowanie kompleksowego systemu informatycznego obejmującego aplikacje mobilną, webową oraz zintegrowany z nimi serwer API. System ma za zadanie uprościć proces rezerwacji wizyt w salonach fryzjerskich, zwiększyć dostępność usług, a także usprawnić zarządzanie wizytami zarówno z perspektywy klienta, jak i pracownika. Ponadto celem systemu jest również przyciąganie nowych klientów i budowanie ich zaufania poprzez umożliwienie oceny zrealizowanych wizyt, przeglądanie ocen innych klientów i udział w programie lojalnościowym. Korzyści z wdrożenia niniejszego rozwiązania miałby także właściciel salonu. Do jego dyspozycji są narzędzia do zarządzania ofertą i pracownikami oraz do generowania raportów z działalności salonu.

## 2  PRACE NAD PROJEKTEM

### 2.1  Analiza istniejących rozwiązań

Jednym z najpopularniejszych rozwiązań na rynku zajmującym się obsługą branży usługowej jest aplikacja *Booksy*, która oferuje wachlarz funkcji rezerwacyjnych, takich jak: podgląd dostępnych terminów, filtrowanie usług oraz oceny klientów. *Booksy* umożliwia również zarządzanie harmonogramem pracowników i oferuje opcje płatności online, co czyni ją wszechstronnym narzędziem dla właścicieli salonów i ich klientów. Technologicznie aplikacja bazuje na rozbudowanej infrastrukturze chmurowej, co zapewnia skalowalność i wysoką dostępność usługi.

Mimo swojej popularności, *Booksy* wiąże się z istotnymi ograniczeniami, które otwierają przestrzeń dla alternatywnych rozwiązań. Największym wyzwaniem dla właścicieli salonów jest wysoka prowizja pobierana od każdej pierwszej wizyty, a także miesięczna subskrypcja. Aplikacja nie oferuje również wbudowanego programu lojalnościowego, który mógłby wzmacniać długoterminowe relacje z klientami. Nasz

model biznesowy zakłada jednorazową płatność za aplikację, co długoterminowo obniża koszty użytkowania dla właścicieli salonów. Wprowadzamy również moduł lojalnościowy, który umożliwia zbieranie pieczątek i nagradzanie stałych klientów zniżkami na kolejne wizyty. Dzięki możliwości dostosowania systemu do indywidualnych potrzeb konkretnej sieci salonów, nasze rozwiązanie staje się bardziej elastyczne i spersonalizowane w porównaniu z uniwersalnym podejściem aplikacji takich jak *Booksy*.

## 2.2 Technologie i narzędzia

Aplikację mobilną zdecydowaliśmy się zaimplementować w nowoczesnym frameworku **SwiftUI** [3] [4], co pozwoliło na pełne wykorzystanie ekosystemu iOS, zapewniając intuicyjny interfejs użytkownika oraz doskonałą wydajność na urządzeniach Apple. **Angular** [1] z **TypeScriptem** [5] został wybrany do napisania aplikacji webowej ze względu na jego popularność, elastyczność oraz bogaty ekosystem bibliotek wspierających dynamiczne tworzenie interfejsów użytkownika. Backend oparto na **Java Spring** [2], ponieważ framework ten jest sprawdzony w budowie skalowalnych i wydajnych serwerów API, a także dobrze współpracuje z **MySQL** – bazą danych, która została wdrożona w chmurze **Microsoft Azure**. Wybór tej platformy hostingowej wynikał z jej niezawodności, skalowalności oraz bogatej oferty integracji z innymi usługami.

Wsparcie pracy zespołu zapewniały narzędzia takie jak **GitHub** do kontroli wersji, **Jira** do planowania sprintów i śledzenia postępów prac, **Canva** do stworzenia projektu graficznego logo i elementów wizualnych, a także **Swagger** do dokumentacji API, co znacznie ułatwiło współpracę między zespołem frontendowym a backendowym.

## 2.3 Ograniczenia czasowe i zasoby

Zespół projektowy składał się z czterech osób, z których każda miała jasno przypisane role: jedna osoba odpowiadała za aplikację mobilną, jedna za webową, a dwie za backend i bazę danych.

Projekt był realizowany w podejściu zwinnym **Scrum**, podzielony na osiem jednotygodniowych sprintów w ramach dwóch miesięcy. Każdy sprint kończył się wdrożeniem kolejnych funkcjonalności, takich jak moduł logowania, rezerwacji, oceny wizyt, przeglądania historii wizyt czy panel właściciela. Dzięki temu możliwe było stopniowe dostarczanie funkcji i bieżące wprowadzanie poprawek na podstawie informacji zwrotnych zespołu. Pomimo napiętego harmonogramu, podejście iteracyjne pozwoliło na skuteczny przebieg projektu.

## 2.4 Napotkane problemy

Podczas realizacji projektu zespół napotkał kilka istotnych problemów:

1. Problemy z synchronizacją API z frontendem - w jednym ze sprintów backend był gotowy później niż aplikacje klienckie, co opóźniło testowanie i integrację funkcji rezerwacji wizyt.

2. Wybór platformy hostingowej - pierwotnie wybrana platforma przetwarzała żądania w czasie liczonym w sekundach, co jest niedopuszczalne dla tego typu aplikacji. Dane ładowały się na ekranie na tyle długo, że było to frustrujące.

3. Dopracowanie interfejsu użytkownika, tak aby był estetyczny, intuicyjny i spójny w obydwu aplikacjach, ale zarazem umożliwiał łatwe korzystanie z dostępnych funkcji.

4. Nieprecyzyjna definicja bazy danych - w dalszej części projektu zaszła potrzeba dodania kilku nowych atrybutów do bazy danych, co wymagało modyfikacji istniejących tabel i wiązało się z dodatkowymi pracami związanymi z migracjami bazy, testowaniem i integracją nowych danych.

5. Zmiana wymagania dotyczącego liczby usług podczas wizyty - w początkowych fazach projektu postanowiliśmy zmienić założenie na temat maksymalnej liczby usług, które można wybrać do jednej wizyty (z jednej do trzech), co skutkowało koniecznością zmian w bazie danych i interfejsie użytkownika.

# 3 WYNIKI

## 3.1 Zaimplementowane funkcjonalności

Zespołowi udało się zaimplementować wszystkie zaplanowane funkcjonalności.

- Dla klienta

- Rejestracja i logowanie.
- Przeglądanie ofert salonów.
- Wybór usług i rezerwacja wizyty.
- Zarządzanie wizytami i ich przeglądanie.
- Wystawianie ocen i przeglądanie ocen innych klientów.
- Program lojalnościowy.

- Dla pracownika
    - Logowanie.
    - Umawianie wizyt dla klientów
    - Przeglądanie swoich wizyt i zarządzanie nimi.

- Dla właściciela
    - Zarządzanie salonami, ofertą usług i pracownikami.
    - Generowanie raportów z działalności salonu.

## 3.2  Zrealizowane cele

Projekt zrealizował następujące cele:

- Biznesowe:
    - Uproszczenie procesu rezerwacji wizyt, co przekłada się na wyższą satysfakcję klientów i zwiększenie liczby wizyt w salonach.
    - Obniżenie kosztów operacyjnych dla właścicieli salonów w porównaniu z konkurencyjnymi rozwiązaniami, eliminując konieczność płacenia prowizji od wizyt i comiesięcznej subskrypcji.
    - Zwiększenie lojalności klientów dzięki wdrożeniu programu lojalnościowego.
    - Wsparcie strategii marketingowej salonów poprzez dostęp do danych analitycznych.

- Techniczne:
    - Integracja między aplikacjami mobilną i webową dzięki solidnemu API opartemu na Java Spring, co umożliwia szybkie i stabilne przetwarzanie żądań.
    - Optymalizacja bazy danych zapewniająca średni czas odpowiedzi na żądania wynoszący 120 ms.
    - Intuicyjne interfejsy użytkownika, dostosowane zarówno do urządzeń mobilnych, jak i komputerów.

## 3.3  Zastosowanie praktyczne

Nasze rozwiązanie znajduje praktyczne zastosowanie w codziennej działalności salonów fryzjerskich, upraszczając proces rezerwacji i zarządzania wizytami. Dzięki aplikacji mobilnej i webowej klienci mogą wygodnie rezerwować wizyty, przeglądać dostępne terminy oraz oceniać usługi, co zwiększa ich satysfakcję i lojalność. System lojalnościowy motywuje stałych klientów do powrotu, oferując zniżki za regularne wizyty. Pracownicy salonu zyskują narzędzia do lepszego planowania swojego czasu pracy, a właściciele mogą monitorować działalność salonu, analizować popularność usług i generować raporty, co wspiera podejmowanie strategicznych decyzji. Eliminacja kosztownych prowizji od rezerwacji obniża koszty operacyjne, co szczególnie korzystne jest dla mniejszych przedsiębiorstw. System jest skalowalny i łatwy do wdrożenia, co czyni go praktycznym rozwiązaniem zarówno dla pojedynczych salonów, jak i dużych sieci, znacząco zwiększając ich konkurencyjność i efektywność.

# 4 WNIOSKI

## 4.1 Konkluzje

Projekt zakończył się sukcesem, dostarczając rozwiązanie, które odpowiada na kluczowe potrzeby rynku fryzjerskiego. System nie tylko usprawnia rezerwację wizyt, ale także integruje funkcje wspierające zarządzanie salonem i budowanie relacji z klientami. Dzięki elastyczności wdrożenia oraz możliwości personalizacji, rozwiązanie spełnia wymagania zarówno małych, jak i dużych przedsiębiorstw, a jego modułowa konstrukcja ułatwia rozwój w przyszłości.

Najważniejszym osiągnięciem jest stabilność i funkcjonalność systemu, który w testach wykazał się wysoką wydajnością i niezawodnością. Aplikacja w praktyce upraszcza procesy operacyjne salonów, jednocześnie podnosząc komfort klientów i wspierając ich zaangażowanie. Opracowany system stanowi przykład nowoczesnego podejścia do cyfryzacji branży usługowej, oferując praktyczne korzyści zarówno użytkownikom, jak i właścicielom biznesów.

## 4.2 Kierunki rozwoju

W przyszłości projekt może zostać rozwinięty o szereg funkcji, które jeszcze bardziej zwiększą jego atrakcyjność i użyteczność. Jednym z kierunków rozwoju są funkcje społecznościowe, między innymi możliwość obserwowania salonów, pracowników czy innych użytkowników oraz dzielenia się opiniami czy inspiracjami stylizacyjnymi w formie zdjęć lub komentarzy. Dzięki temu system mógłby pełnić rolę platformy społecznościowej, budując zaangażowaną społeczność wokół usług fryzjerskich.

Kolejną propozycją jest większa personalizacja doświadczeń klienta. Możliwość tworzenia indywidualnych profili z preferencjami dotyczącymi usług, ulubionych fryzjerów czy nawet zapisanej historii stylizacji pozwoliłaby użytkownikom na jeszcze wygodniejsze korzystanie z aplikacji. Dodatkowo system mógłby rekomendować usługi na podstawie wcześniejszych wyborów klienta, oferując np. promocje dopasowane do jego preferencji.

Nowatorskim rozwiązaniem mogłaby być również integracja z systemami rozszerzonej rzeczywistości, umożliwiająca klientom wirtualne przymierzanie fryzur czy kolorów włosów przed wizytą. Tego rodzaju funkcja nie tylko zwiększyłaby satysfakcję klientów, ale także zmniejszyłaby ryzyko niezadowolenia z efektów końcowych. Rozszerzenie projektu o te funkcje nie tylko podniosłoby jego konkurencyjność, ale również wpłynęło na zwiększenie lojalności klientów i zainteresowanie nowoczesnym podejściem do usług fryzjerskich.

## 4.3 Podziękowania

Chcielibyśmy serdecznie podziękować opiekunowi naszego zespołu, Pani mgr Hannie Mazur, za nieocenione wsparcie, zaangażowanie oraz cenne wskazówki, które towarzyszyły nam na każdym etapie realizacji projektu. Jesteśmy wdzięczni za poświęcony czas oraz wsparcie merytoryczne, znacząco wpłynęło to na nasz rozwój jako zespołu i przyszłych specjalistów.

# LITERATURA

[1] Angular framework documentation. https://angular.dev/overview, 2024. Dostęp: listopad 2024.

[2] Spring framework documentation. https://docs.spring.io/spring-framework/reference/index.html, 2024. Dostęp: listopad 2024.

[3] Swift documentation. https://www.swift.org/documentation/, 2024. Dostęp: listopad 2024.

[4] Swiftui framework documentation. https://developer.apple.com/documentation/swiftui/, 2024. Dostęp: listopad 2024.

[5] Typescript documentation. https://www.typescriptlang.org/docs/, 2024. Dostęp: listopad 2024.
