# OpenReview forum: "System rezerwacji wizyt w salonie fryzjerskim"
_pwr.edu.pl/Wrocław_University_of_Science_and_Technology/2024/ZPI_Day — Wrocław University of Science and Technology 2024 ZPI Day Submission_

### Official Review · Reviewer_ngcB · 2024-12-04
**Poprawnie napisany artykuł, oczekiwano bardziej szczegółowego opisu implementacji**

**Confidence:** 5
**Significance Of Results:** 4
**Overall Quality:** 4

**Compliance With Template:**

4: High Quality – The article contains all the required sections, which are well-written and substantively correct, although minor errors or shortcomings may be present. The overall structure is clear and coherent.

**Description Of Results:**

3: Average Quality – The results are described with moderate detail. Some examples or evaluation elements are present but insufficiently developed or incomplete.

**Feedback On Consistency:**

+ Artykuł napisany jest ładnym językiem. Struktura dokumentu jest poprawna. Spójnie stosowano czas przeszły dla prac zrealizowanych oraz przyszły dla planowanych rozwiązań.

- Brakuje niestety choćby najmniejszego obrazka przedstawiającego kluczowe elementy systemu. Jego załączenie pozwoliłoby ocenić szatę graficzną aplikacji webowej. Dałoby podstawowy wgląd w UI/UX projektu webowego.

**Potential For Development:**

+ Aktualny poziom funkcjonalny systemu jest adekwatny do czasu przeznaczonego na jego opracowanie. Autorzy twórczo i poprawnie wyznaczyli dalsze kierunki rozwoju. Szczególnie obiecującą funkcjonalnością wydaje się być możliwość wirtualnego doboru fryzury.

**Project Nature Evaluation:**

+ Projekt całkowicie wpisuje się w naturę pracy inżynierskiej. Ostrożnie podchodziłbym do wypowiedzi nie technicznych, a raczej biznesowych dotyczących preferencji funkcjonalności w systemie. Wysoka prowizja Booksy ma charakter ograniczenia ekonomicznego a nie inżynierskiego. Można by powiedzieć, że gdyby Booksy bez zmian w kodzie stało się darmowe, to przewaga opracowania autorów znika. Lepiej jest porównywać różnice technologiczne, takie jako UI/UX, prędkość działania, braki funkcjonalności, dostęp za pomocą zróżnicowanych kanałów, etc. Są to aspekty bardziej odpowiadające inżynierii oprogramowania.

+ Technologie wykorzystane do opracowania aplikacji są uzasadnione. Choć pewną obawę biznesową budzi wykorzystanie SwiftUI do napisania aplikacji mobilnej. Technologia ta pokrywa tylko smartfony Apple iOS. A co z Androidem?

- Sporym minusem jest zbyt ogólnikowy rozdział 3, zatytułowany "Wyniki". Wylistowanie funkcjonalności wydaje się być zbyt krótkim opisem najważniejszych aspektów inżynierskich pracy. Oczekiwałbym, aby właśnie ten rozdział dostarczał szerokiej informacji o projekcie, ponieważ jest on bezpośrednim dowodem na posiadane umiejętności inżynierskie autorów. Cała reszta jest raczej tłem względem osiągniętych rezultatów.

**Technical Language Precision:**

4: High Quality – The language is appropriate for a technical report. Terminology is used correctly, and statements are precise, with only minor shortcomings that do not affect the overall clarity.

---

### Official Review · Reviewer_Tvcc · 2024-12-04
**System rezerwacji wizyt w salonie fryzjerskim - ocena ogólna (w skali 1-5): 5**

**Confidence:** 5
**Significance Of Results:** 5
**Overall Quality:** 5

**Compliance With Template:**

5: Very High Quality – The article contains all the required sections, which are written in a very detailed, clear, and error-free manner. The structure is professional and meets expectations, and the content adheres to the highest substantive and formal standards.

**Description Of Results:**

5: Very High Quality – The results are described in detail, clearly and comprehensively, supported by thorough evaluation, analysis, and convincing usage examples. The description meets the highest substantive standards.

**Feedback On Consistency:**

Praca ogólnie poprawna. Zauważone błędy:
- żargon (np. stworzenie)
- przed nazwami własnymi brak określeń typu język, baza danych itp.
- błędy stylistyczne, gramatyczne, interpunkcyjne
- niejednolity format wypunktowań
- pozostawianie "wdów" czyli pojedynczych słów w ostatnim wierszu akapitu
- pozostawianie "sierot" czyli pojedynczej litery na końcu wiersza, np. w tekście "niezadowlenia z efektów końcowych"

**Potential For Development:**

Opisano kierunki rozwoju

**Project Nature Evaluation:**

Zespołowy projekt inżynierski

**Technical Language Precision:**

5: Very High Quality – The language is entirely appropriate for a technical report. All terms are used correctly and precisely, and the style is professional, clear, and coherent, without any errors or ambiguities.

---

### Official Review · Reviewer_mUP1 · 2024-12-08
**IT project prepared correctly.**

**Confidence:** 5
**Significance Of Results:** 5
**Overall Quality:** 5

**Compliance With Template:**

5: Very High Quality – The article contains all the required sections, which are written in a very detailed, clear, and error-free manner. The structure is professional and meets expectations, and the content adheres to the highest substantive and formal standards.

**Description Of Results:**

5: Very High Quality – The results are described in detail, clearly and comprehensively, supported by thorough evaluation, analysis, and convincing usage examples. The description meets the highest substantive standards.

**Feedback On Consistency:**

The analysis of the problem, presentation of results and conclusions are coherent and logical

**Potential For Development:**

The article indicates possibilities for further work or practical application of its results.

**Project Nature Evaluation:**

Both the level of usability, the applied technical methods and technological solutions have the characteristics of engineering work.

**Technical Language Precision:**

5: Very High Quality – The language is entirely appropriate for a technical report. All terms are used correctly and precisely, and the style is professional, clear, and coherent, without any errors or ambiguities.

---

### Decision · Program_Chairs · 2024-12-10

Accept (Poster)